# Bactericidal effect of far ultraviolet-C irradiation at 222 nm against bacterial peritonitis

**Kosuke Sugiyama**[1], **Kiyotaka Kurachi**[1]*, **Masaki Sano**[1], **Kyota Tatsuta**[1], **Tadahiro Kojima**[1], **Toshiya Akai**[1], **Katsunori Suzuki**[1], **Kakeru Torii**[1], **Mayu Sakata**[1], **Yoshifumi Morita**[1], **Hirotoshi Kikuchi**[1], **Yoshihiro Hiramatsu**[1,2], **Yohei Kumabe**[3], **Keisuke Oe**[3], **Tomoaki Fukui**[3], **Rena Kaigome**[4], **Masahiro Sasaki**[4], **Toru Koi**[4], **Hiroyuki Ohashi**[4], **Tetsuro Suzuki**[5], **Ryosuke Kuroda**[3], **Hiroya Takeuchi**[1]

1 Department of Surgery, Hamamatsu University School of Medicine, Hamamatsu, Shizuoka, Japan,
2 Department of Perioperative Functioning Care and Support, Hamamatsu University School of Medicine, Hamamatsu, Shizuoka, Japan, 3 Department of Orthopaedic Surgery, Kobe University Graduate School of Medicine, Kobe, Hyogo, Japan, 4 Ushio Inc., Chiyoda-ku, Tokyo, Japan, 5 Department of Microbiology and Immunology, Hamamatsu University School of Medicine, Hamamatsu, Shizuoka, Japan

* kurachi1@hama-med.ac.jp

**Data Availability Statement:** All relevant data are within the manuscript and its Supporting Information files. We have uploaded a description of the data set as Supporting Information files to Figshare.(DOI: 10.6084/m9.figshare.26196038).

## Abstract

Far ultraviolet-C irradiation at 222 nm has potent bactericidal effects against severe infections such as peritonitis, with minimal cytotoxicity. Bacterial peritonitis due to bowel perforation is a serious condition with high mortality despite current treatments. This study investigated the safety and efficacy of intraperitoneal far ultraviolet-C irradiation at 222 nm. *In vitro* experiments optimized the fluid conditions for bacterial or protein concentrations prior to in vivo evaluation. In vivo efficacy was assessed in a rat peritonitis model induced by *Escherichia coli*, measuring intra-abdominal bacterial concentration, blood cytokine levels, and mortality rates. Safety was evaluated by analyzing cyclobutane pyrimidine dimers as markers of DNA damage in five abdominal organs: stomach, small intestine, colon, liver, and spleen. Statistical analyses employed parametric methods for normally distributed data and non-parametric methods for data without normality. Optimal *in vitro* conditions included $10^6$ CFU/mL bacteria, 0.5 mW/cm$^2$ irradiation, and $10^{-3}$ mg/mL protein. In the rat model, far ultraviolet-C irradiation at 222 nm significantly decreased intra-abdominal bacteria, reduced blood cytokines (interleukin-1β and interleukin-6), and elevated survival rates from 20% to 60%, compared to lavage alone. The formation of cyclobutane pyrimidine dimers was significantly lower with 222 nm irradiation than with 254 nm, suggesting reduced DNA damage. These findings indicate that far ultraviolet-C irradiation at 222 nm, when combined with lavage, represents a promising therapeutic strategy for bacterial peritonitis, providing effective bacterial reduction and a favorable safety profile. Further research is needed to verify these findings and investigate long-term safety and potential clinical applications.

**Funding:** The author(s) received no specific funding for this work.

**Competing interests:** Because this research required specialized experimental equipment, it was started as a joint research project between Kobe University, Ushio Inc., and Hamamatsu University of Medicine. This experiment was performed at Hamamatsu University School of Medicine. The staff of Ushio Inc. and Kobe University lent specialized equipment used in another study to the Hamamatsu University School of Medicine, evaluated its appropriate use, assessed the validity of the methods and results, and confirmed the reproducibility of some data. Thus, the funders had no role in data collection and analysis. Based on the result of this research, Hamamatsu University School of Medicine have received funds from Ushio Inc. associated with this collaborative project. Kobe University School of Medicine and Hamamatsu University School of Medicine are funded by Ushio Inc. for collaborating research. RK, MS, TK, and HO have received support in the form of salaries from Ushio Inc. This does not alter our adherence to PLOS ONE policies on sharing data and materials

## Introduction

Acute peritonitis, caused by gastrointestinal perforation, surgical site infection, or spontaneous bacterial infection, is a severe disease with high morbidity and mortality. Despite available multimodal therapies, the 90-day mortality rate of acute diffuse peritonitis remains over 10% [1]. During laparotomy, the infected area is resected, and intraperitoneal lavage with saline is performed to reduce bacterial concentration and control the infection [2]. These surgical procedures, followed by antibiotic administration, are the established treatments for acute peritonitis [2–9]. However, while extensive lavage decreases intra-abdominal bacterial concentration, it is challenging to completely eradicate the infection [1, 10]. In 50% of cases during elective colorectal surgery, bacterial concentrations remain below $10^6$ CFU/mL even without visible contamination after lavage [10]. The residual bacteria may lead to postoperative infections, as the risk escalates proportionally with bacterial levels. Persistent infection from ascites, even in the absence of visible contamination, presents a significant challenge in managing bacterial peritonitis. This highlights the critical need for bacterial reduction in the treatment of severe bacterial peritonitis, regardless of the apparent clarity of the lavage fluid.

Therefore, novel treatments for intra-abdominal infections are required. Some studies have been conducted to add antibiotics or disinfectants to lavage solutions for the treatment of peritonitis [9, 11–17]. However, it has also been indicated that these additives might exacerbate systematic inflammation [9, 18–20]. Intraperitoneal lavage with large-volume saline alone cannot reduce postoperative mortality and complications of acute peritonitis. Additional bactericidal treatments beyond lavage might be necessary for the surgical treatment of acute peritonitis.

The bactericidal efficacy of ultraviolet-C (UV-C) irradiation is well-established. Specifically, UV-C irradiation within the 200–280 nm range demonstrates strong bactericidal properties [21]. Notably, UV-C at 254 nm is frequently employed in bactericidal devices due to its potent bactericidal effects; however, its clinical application is limited by significant cytotoxicity, including DNA damage leading to eye keratitis and malignant skin tumors [22–24].

Longer wavelengths of UV-C, such as 254 nm, have higher cytotoxicity, while shorter wavelengths can form ozone and free radicals, causing biological damage [25]. Recently, far UV-C, defined as wavelengths between 200–230 nm, has garnered interest for its comparable bactericidal effect and increased safety for mammalian tissues relative to traditional UV-C. Our study particularly concentrated on far UV-C irradiation at 222 nm, as devices for this wavelength are commercially available and easily accessible. Furthermore, far UV-C at 222 nm offers advantages with its effective bactericidal properties against skin surface infections and its reduced cytotoxicity to the skin, subcutaneous tissue, and eyes compared to UV-C at 254 nm [26–29]. Although devices with variable wavelengths are available, our focus was specifically on 222 nm due to its well-documented efficacy and safety in treating superficial infections [26–29].

A previous clinical trial showed that far UV-C irradiation at 222 nm of the skin at 500 mJ/cm² was safe for use in humans [30]. Furthermore, far UV-C irradiation at 222 nm has an equivalent bactericidal effect and less cytotoxicity compared with UV-C irradiation at 254 nm [27] and may, therefore, be a novel treatment method for intraperitoneal infection.

The bactericidal effects and cytotoxicity of UV-C irradiation depend on the relationship between the absorption coefficient and cell size [31]; therefore, they differ among bacterial species and cell types. Moreover, the bactericidal effect of UV-C irradiation is influenced by several factors, such as bacterial concentration, irradiance, fluid protein concentration, and transmittance rate [32–34]. Previous studies have focused on the therapeutic effect of far UV-C irradiation at 222 nm on intra-abdominal infections [26, 27]; however, the effects of far UV-C irradiation at 222 nm on contaminated ascites containing bacteria and proteins remain

unknown. The objective of this study was to evaluate the bactericidal effect and cytotoxicity of far UV-C irradiation at 222 nm in the intra-abdominal cavity both *in vitro* and *in vivo*.

## Materials and methods

### Experimental design

*In vitro* experiments were conducted to optimize the condition for the *in vivo* investigation of the bactericidal effect of far UV-C irradiation at 222 nm; the *in vivo* study comprised three experiments. First, we evaluated the efficacy of far UV-C irradiation at 222 nm for peritonitis using two protocols: acute phase (1 week) and hyper-acute phase (3 h) following the induction of peritonitis. Second, we evaluated the damage to the abdominal organs induced by far UV-C irradiation at 222 nm.

### Study approval

All animal procedures and experimental protocols were performed in accordance with the ethical standards of the Animal Care and Use Committee of the Hamamatsu University School of Medicine (approval number 2022008). The study protocols complied with the Animal Research Reporting of In Vivo Experiments guidelines.

### Bacterial culture and measurement

*Escherichia coli (E. coli)* was selected due to its significant role in colorectal perforation and its ability to induce systemic inflammation through cytokine release and endotoxin production. *E. coli* (Migula) Castellani and Chalmers (ATCC 25922) was obtained from the American Type Culture Collection (Manassas, Virginia, US) and cultured in tryptic soy broth (BD #211825, BD Difco US Co., Ltd.) at 37°C with SLI-1200 (Tokyo Rikakikai Co., Ltd., Tokyo Japan). To determine the bacterial concentration, the colony-forming units (CFU)/mL were measured after streak-plate culture and diluted $10–10^{10}$-fold with sterilized saline. From each diluted solution, 100 μL was added to a deoxycholate-agar plate, streaked, and incubated for 24–48 h at 37°C.

### UV-C irradiation

A 222 nm-emitting Care222Ⓡ device (Ushio Inc. Tokyo, Japan) was used for far UV-C irradiation at 222 nm, as previously described [29, 31, 35]. This device is a filtered Krypton-Chloride (KrCl) excimer far UV-C, constructed with KrCl excimer lamps and an irradiator with a filter that filters out the UV-C region. The irradiance emitted by the 222-nm light was measured using an S-172/UIT250 accumulated UV meter (Ushio Inc.). UV-C irradiation (254 nm) was conducted with a low-pressure mercury lamp (SUV-4, AS ONE, Osaka, Japan). The measurement of UV-C irradiance at 254 nm was performed similarly to that of far UV-C at 222 nm. The irradiance of far UV-C at 222 nm inversely correlates with the square of the distance from the irradiation source; consequently, the greater the distance, the lower the irradiance. Moreover, sufficient irradiance levels are maintained only over short distances. The irradiation dose is calculated based on irradiance and exposure time, using the formula below. We categorized the irradiation dose as either low or high, based on the threshold limit values for 222 nm UV-C.

Irradiation dose $([mJ]/[cm^2])$ = Irradiance $(mW/cm^2) \times$ Time (s)

For this study, consistent irradiation doses were required. Following the aforementioned formula, we adjusted the irradiance and distance in a peritonitis rat model. The settings were 0.5 mW/cm$^2$ at a 20 cm distance for 40 s, and 0.25 mW/cm$^2$ at a 40 cm distance for 1000 s.

Achieving an irradiance greater than 0.5 mW/cm$^2$ was challenging due to the proximity constraints between the light source and the rat model, as was maintaining less than 0.25 mW/cm$^2$ over extended distances due to insufficient irradiance.

## Bactericidal effect of far UV-C irradiation at 222 nm *in vitro*

On a clean bench, *E. coli* dilutions in sterile saline were dispensed into microwells at a depth of 1 cm. According to previously reported experimental protocols [36, 37], the doses of far UV-C irradiation were divided into low dose (LD, 20 mJ/cm$^2$, 0.5 mW/cm$^2$, 40 s) and high dose (HD, 500 mJ/cm$^2$, 0.5 mW/cm$^2$, 1000 s). The *E. coli* solution was exposed to far UV-C irradiation at 222 nm at the above two doses. The microwells were shaken gently during irradiation. After irradiation, the bacterial concentrations were measured and compared with those of the negative control (NC, not irradiated).

A reduction of > 2 log CFU/mL, which indicates > 99% reduction, was considered to denote a confirmed bactericidal effect [26]. We also confirmed the influence of irradiance on the bactericidal effect of far UV-C irradiation at 222 nm. The irradiance of far UV-C at 222 nm was set at 0.25 (LD: 80 sec, HD: 2000 sec) or 0.5 mW/cm$^2$ (LD: 40 sec, HD: 1000 sec). After irradiation, the bacterial concentrations were measured and compared among the three groups (NC, LD, and HD).

We evaluated the bactericidal effect of far UV-C irradiation at 222 nm in protein-containing solutions with soy broth as pseudo-ascites, performed multiple dilutions of soy broth with saline, and adjusted the protein concentration from $1.0 \times 10^{-3}$ to 1.5 mg/mL. *E. coli* dilutions in soy broth solutions were also dispensed into microwells at a depth of 1 cm and exposed to far UV-C irradiation at 222 nm with an irradiance of 0.5 mW/cm$^2$ (LD: 40 sec, HD: 1000 sec).

Further, we evaluated the influence of protein levels and solution depth on the transmittance rate of far UV-C irradiation at 222 nm. The concentrations of soy broth solution were adjusted to $1.0 \times 10^{-3}$, $1.5 \times 10^{-3}$, and $1.5 \times 10^{-2}$ mg/mL, respectively, while the irradiance was set at 0.5 mW/cm$^2$ using a luminometer positioned at the bottom of the quartz cells. Protein solution was introduced to quartz cells at depths of 1.5, 3.0, and 4.5 cm. We then irradiated the quartz cuvettes with far UV-C at 222 nm, shielding them from ambient light. The transmittance rate was determined by comparing the ratio of far UV-C irradiance at 222 nm at depths of 1.5, 3.0, and 4.5 cm to that of the blank quartz cuvettes, with n = 5 for each soy broth concentration.

## Rat peritonitis model

For the *in vivo* experiments, 16–18-week-old male Sprague-Dawley rats (Japan SLC, Inc., Hamamatsu, Japan; 450–500 g) were used. All rats were intraperitoneally anesthetized with 0.4 mg/kg medetomidine, 2.0 mg/kg midazolam, and 5.0 mg/kg butorphanol before the surgical procedures. The rats were divided into two groups: control and peritonitis (n = 5 per group). In the peritonitis group, 2 mL of *E. coli* solution adjusted to $10^8$ CFU/mL with saline was injected into the lower right abdomen [38]. On the other hand, 2 mL of saline was injected in the control group. As physical findings at the time of peritonitis induction, rats with peritonitis tend to exhibit behavioral changes such as reluctance to stretch the abdomen and immobility. Additionally, as objective indicators to evaluate peritonitis, we measured the bacterial concentration in the ascites and blood cytokines to confirm their increase. Three hours after injection, we measured the ascitic bacterial and serum cytokine concentrations and compared the control and peritonitis groups (Fig 1A).

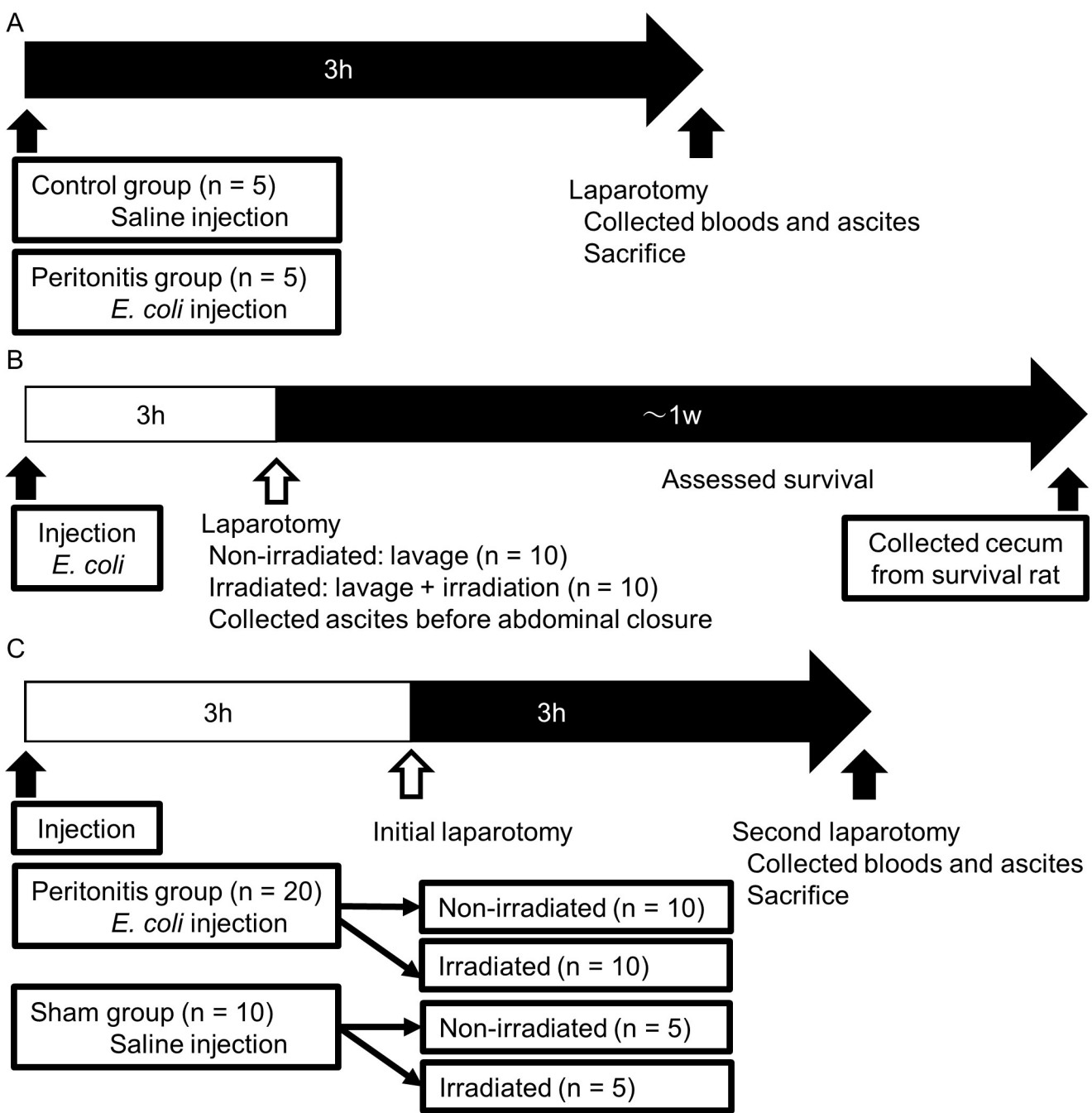

**Fig 1. *In Vivo* experimental protocols to evaluate the bactericidal effect of far UV-C irradiation on peritonitis.** (A) Sample collection of rat peritonitis model. Ten rats were divided into control and peritonitis groups (n = 5 in each). Three hours after the injection of *E. coli* solution or saline, the laparotomy was performed, and blood samples and ascites were collected. (B) Evaluation in the acute phase. Twenty rats were divided into non-irradiated and irradiated groups (n = 10 in each). Three hours after the injection of the *E. coli* solution, a laparotomy was performed. Lavage was performed in the non-irradiated group, and lavage and irradiation were performed in the irradiated group. After these procedures, the lavage solution was collected. The survival rate was evaluated, and the surviving rats were sacrificed 1 week after the procedure to obtain the cecum. (C) Evaluation in the hyper-acute phase. Thirty rats were divided into peritonitis (n = 20) and sham groups (n = 10). Each group was divided into two sub-groups, namely non-irradiated and irradiated. Three hours after the injection of *E. coli* solution or saline, an initial laparotomy was performed. Lavage was performed in the non-irradiated group, and lavage and irradiation were performed in the irradiated group. Three hours after the procedures, the second laparotomy was performed, and blood samples and ascites were collected.

## Bactericidal effect of far UV-C irradiation on peritonitis

We used two protocols to evaluate the bactericidal effect of far UV-C irradiation at 222 nm on acute and hyper-acute peritonitis using a rat model. We performed laparotomy once in the acute phase protocol and twice in the hyper-acute phase protocol.

The evaluation protocol during the acute phase is illustrated in Fig 1B. Twenty Sprague-Dawley rats were randomly divided into two groups (non-irradiated and irradiated, n = 10 in each group). *E. coli* solution was injected into all rats as described above. Three hours after the injection of the *E. coli* solution, a laparotomy was performed. In the non-irradiated group, intra-abdominal lavage was performed twice with 50 mL of saline; however, irradiation was not conducted. Conversely, in the irradiated group, intra-abdominal lavage was also performed twice with 50 mL of saline, and far UV-C irradiation at 222 nm with an intensity of 20 mJ/cm$^2$ (0.5 mJ/cm$^2$ for 40 s) was administered during the second lavage. In both groups, 5 mL of a second lavage solution was collected immediately before abdominal closure. Subsequently, the survival rate was assessed, and the surviving rats were euthanized with an overdose of intraperitoneal anesthetics (1.2 mg/kg medetomidine, 6.0 mg/kg midazolam, and 15.0 mg/kg butorphanol) 1 week post-laparotomy to harvest the cecum and to assess DNA damage induced by far UV-C irradiation through histopathological evaluation. Notably, the cecum, being large and susceptible to irradiation, was included in the analysis.

The evaluation protocol for the hyper-acute phase is illustrated in Fig 1C. Thirty Sprague-Dawley rats were randomly divided into two groups: peritonitis (n = 20) and sham (n = 10). *E. coli* solution was injected into the rats in the peritonitis group, while saline was injected into those in the sham group. The rats in each group were divided into two sub-groups, namely irradiated and non-irradiated. Three hours after the injection of *E. coli* solution or saline, the initial laparotomy was performed. In the non-irradiated group, intra-abdominal lavage was performed twice with 50 mL of saline, while irradiation was not performed. In the irradiated group, intra-abdominal lavage was performed twice with 50 mL of saline, and far UV-C irradiation at 222 nm at 20 mJ/cm$^2$ (0.5 mJ/cm$^2$ for 40 s) was performed during the second lavage. The abdomen was closed after the procedure. Three hours after the initial laparotomy, a second one was performed. Blood samples were collected from the portal vein. The volume of the ascitic fluid was too small to be collected and analyzed. The lavage solution was collected after dispensing 10 mL of saline into the abdominal cavity and gently stirring.

In this study, antibiotics were not administered in any of the protocols. However, antibiotic administration is the established treatment for acute peritonitis. This may modify and obscure the therapeutic effects of intra-abdominal lavage and UVC treatment.

## Analysis of lavage solution and blood samples

The bacterial concentration (CFU/mL) in the lavage solution was determined after streak-plate culturing and diluted 10–10$^{10}$-fold with sterilized saline.

The protein concentration of the lavage solution was measured. The lavage solution was added into a quartz cuvette and measured with spectrophotometry using an analyzer (DS-11+, Denovix Inc., Wilmington, DE, US) and the ProteinA280 software (Thermo Fisher Scientific Inc., Commonwealth of Massachusetts, US).

Within 30 min of collection, the blood samples were centrifuged at 3,000 rpm for 5 min, and serum was collected and stored frozen at -80°C. Based on the datasheet, the interleukin (IL)-1β, IL-6, and tumor necrosis factor (TNF-α) concentrations were measured using the Luminex® Discovery Assay Rat Premixed Multi-Analyte Kit (R&D Systems, Inc. Minneapolis, MN, US). The analysis was performed by Filgen, Inc. (Nagoya, Japan) using a Bio-Plex

(TM) 200 System (Bio-Rad Laboratories Inc., California, US), and the analysis software was Bio-Plex Manager (TM) Software Version 6.1 (Bio-Rad Laboratories, Inc.).

## Evaluation of cytotoxicity in the abdominal tissues by UV-C irradiation

Twenty rats were randomly divided into four groups: NC, LD, HD, and positive control (PC) (n = 5 per group). Under anesthesia with similar doses at the induction of peritonitis, laparotomy was performed through a ventral midline abdominal incision. The abdominal tissues in the NC group were not exposed to UV-C irradiation. The irradiance was set at $0.5$ mW/cm$^2$, and the abdominal tissues were exposed to far UV-C irradiation at 222 nm at 20 mJ/cm$^2$ ($0.5$ mJ/cm$^2$ for 40 s) in the LD group and 500 mJ/cm$^2$ ($0.5$ mJ/cm$^2$ for 1000 s) in the HD group. In the PC group, the irradiance was set at $0.5$ mW/cm$^2$, and the abdominal tissues were exposed to UV-C irradiation at 254 nm at 75 mJ/cm$^2$ ($0.5$ mJ/cm$^2$ for 150 s). After irradiation, five tissue samples (small intestine, ascending colon, stomach, liver, and spleen) were obtained from each rat, fixed in formalin, and underwent histopathological analysis to evaluate UV-C-induced DNA damage.

## Immunohistochemical staining for cyclobutane pyrimidine dimers

Immunohistochemical (IHC) staining for cyclobutane pyrimidine dimers (CPD) was performed to evaluate UV-C-induced DNA damage [31]. The obtained abdominal tissues were fixed in 4% formaldehyde at 24˚C for 24 h and embedded in paraffin wax. The deparaffinized and rehydrated sliced sections (3 μm-thick) were incubated with CPD antibody (1:1,000, Cosmo Bio Co., Ltd., Tokyo, Japan), diluted with phosphate-buffered saline containing 1% bovine serum albumin, overnight at 4˚C. Histofine Simple Stain MAX Po (M) (Nichirei Bioscience Inc., Tokyo, Japan) was used as a secondary antibody, with incubation for 40 min. Simple Stain aminoethyl carbazole solution (Nichirei Bioscience Inc.) was used for detection. The slides were counterstained with hematoxylin. Images were captured using a BX51 microscope (Olympus, Tokyo, Japan).

Three fields were randomly selected, and the CPD-positive rate and depth were measured. The numbers of CPD-positive and CPD-negative cells were counted. The percentage of CPD-positive cells indicated the CPD-positive rate [31], while the distance from the tissue surface to the deepest CPD-positive cells indicated the CPD-positive depth [39].

## Statistical analyses

The distribution of continuous variables was assessed using the Kolmogorov–Smirnov test. Normality was confirmed for the transmittance rate, CPD-positive rate, and CPD-positive depth, and the data are shown as the mean and standard deviation. Differences between these data were analyzed using parametric analysis: Student's *t*-test between two groups or analysis of variance (ANOVA), followed by post hoc analysis with Tukey's procedure, among more than two groups. Normality was not confirmed for the bacterial, protein, and serum cytokine (IL-1β, IL-6, and TNF-α) concentrations. Data are presented as the median and interquartile range. Differences between these data were analyzed using non-parametric analysis: Mann–Whitney *U* test between two groups or Kruskal-Wallis test, followed by post hoc analysis with the Dunn-Bonferroni procedure among more than two groups.

The survival rates between the irradiated and non-irradiated groups were compared using Kaplan–Meier analysis with the log-rank test. Correlations between the bacterial concentration and each blood cytokine level were analyzed using Pearson's correlation coefficients. A correlation was considered significant at $r > 0.4$ or $< -0.4$, and statistical significance was set at $p < 0.05$. The results were analyzed using SPSS version 28 (IBM Co., Ltd. New York, US).

## Results

### Bactericidal effect of far UV-C irradiation at 222 nm *in vitro*

Through *in vitro* experiments, we optimized the conditions for bacterial concentration, irradiance, and protein concentration for the *in vivo* experiments.

*E. coli* was diluted with saline to a concentration of $10^{5-7}$ CFU/mL. The irradiance of far UV-C at 222 nm was 0.5 mW/cm$^2$, and the NC, LD, and HD groups were compared. The bacterial concentration in the $10^{5-7}$ CFU/mL solution notably decreased after far UV-C irradiation at 222 nm, with initial bacterial concentrations at various levels showing significant changes post-treatment. At $10^7$ CFU/mL, the untreated control (NC) showed $2.0 \times 10^7$ CFU/mL [$1.8 \times 10^7$–$2.2 \times 10^7$], the LD was $3.4 \times 10^5$ CFU/mL [$2.0 \times 10^5$–$1.2 \times 10^6$], and the HD reached $2.5 \times 10^4$ CFU/mL [$7.7 \times 10^3$–$5.5 \times 10^4$]. At $10^6$ CFU/mL, NC was $2.0 \times 10^6$ CFU/mL [$1.8 \times 10^6$–$2.2 \times 10^6$], LD dropped to $1.6 \times 10^2$ CFU/mL [$1.4 \times 10^2$–$1.8 \times 10^2$], and HD was reduced to 0 CFU/mL [0–5.0]. At $10^5$ CFU/mL, NC remained at $2.0 \times 10^5$ CFU/mL [$1.8 \times 10^5$–$2.2 \times 10^5$], while both LD and HD were reduced to 0 CFU/mL [0–$9.0 \times 10^1$] and $0 \times 0$ CFU/mL [0–0], respectively (p < 0.01) (Fig 2A). The bacterial concentration in the $10^{5-7}$ CFU/mL solution significantly decreased in the LD and HD groups compared with the NC group (p < 0.01). Furthermore, in the $10^6$ and $10^7$ CFU/mL solutions, bacterial concentrations were significantly different between the LD and HD groups (p < 0.01). In the $10^7$ CFU/mL solution, the difference between the NC and LD groups and between the LD and HD groups was less than 2 log CFU/mL. Conversely, in the $10^6$ CFU/mL solution, reductions exceeded 2 log CFU/mL between the NC and LD groups and between the LD and HD groups. In the $10^5$ CFU/mL solution, reductions between the NC and LD groups also surpassed 2 log CFU/mL, while no significant difference was observed between the LD and HD groups (p = 0.690). Differences greater than 2 log CFU/mL between the NC and LD groups, and between the LD and HD groups, were exclusively observed in the $10^6$ CFU/mL solution.

The bactericidal effect of far UV-C irradiation at 222 nm, influenced by irradiance, is depicted in Fig 2B. *E. coli* was diluted in saline to a concentration of $10^6$ CFU/mL. Subsequent exposure to far UV-C irradiation at both 0.25 and 0.5 mW/cm$^2$ resulted in a significant reduction in bacterial concentration (0.5 mW/cm$^2$: $10^7$ CFU/mL, NC: $2.0 \times 10^7$ CFU/mL [$1.8 \times 10^6$–$2.2 \times 10^6$], LD: $1.6 \times 10^2$ CFU/mL [$1.4 \times 10^2$–$1.8 \times 10^2$], HD: 0 CFU/mL [0–5.0]; 0.25 mW/cm$^2$: NC: $2.0 \times 10^6$ CFU/mL [$1.8 \times 10^6$–$2.2 \times 10^6$], LD: $5.3 \times 10^5$ CFU/mL [$3.9 \times 10^5$–$7.5 \times 10^5$], HD: $1.4 \times 10^5$ CFU/mL [$1.1 \times 10^5$–$1.7 \times 10^5$], p < 0.01) (Fig 2B). Significant differences in bacterial concentrations were observed between the NC and LD groups and between the LD and HD groups at both 0.25 and 0.5 mW/cm$^2$ (p < 0.01). Specifically, at 0.5 mW/cm$^2$, the bacterial reduction between the NC and LD groups and between the LD and HD groups exceeded 2 log CFU/mL. Similarly, at 0.25 mW/cm$^2$, reductions exceeded 1 log CFU/mL.

We evaluated the bactericidal effect of far UV-C irradiation at 222 nm on protein-containing solutions. *E. coli* was diluted to a concentration of $10^6$ CFU/mL with saline and soy broth solution to adjust the concentration from $1.5 \times 10^{-2}$ to 1.5 mg/mL. The irradiance of far UV-C at 222 nm was set at 0.5 mW/cm$^2$, and the NC, LD, and HD groups were compared (Fig 2C). At a soy broth concentration of 1.5 mg/mL, no significant difference among the NC, LD, and HD groups was observed. At soy broth concentrations of $1.5 \times 10^{-1}$ and $1.5 \times 10^{-2}$ mg/mL, the bacterial concentrations significantly differed among the three groups. In the saline group, the bacterial concentrations were as follows: NC: $2.2 \times 10^6$ CFU/mL [$1.9 \times 10^6$–$2.4 \times 10^6$], LD: $1.6 \times 10^2$ CFU/mL [$1.4 \times 10^2$–$1.8 \times 10^2$], HD: 0 CFU/mL [0–5.0]. At a concentration of 1.5 mg/mL, the results were: NC: $2.0 \times 10^6$ CFU/mL [$1.9 \times 10^6$–$2.4 \times 10^6$], LD: $1.9 \times 10^6$ CFU/mL [$1.4 \times 10^6$–$2.1 \times 10^6$], HD: $1.7 \times 10^6$ CFU/mL [$1.4 \times 10^6$–$2.1 \times 10^6$]. At a concentration of $1.5 \times 10^{-1}$ mg/mL, the results were: NC: $2.0 \times 10^6$ CFU/mL [$1.9 \times 10^6$–$2.4 \times 10^6$], LD: $2.3 \times 10^5$

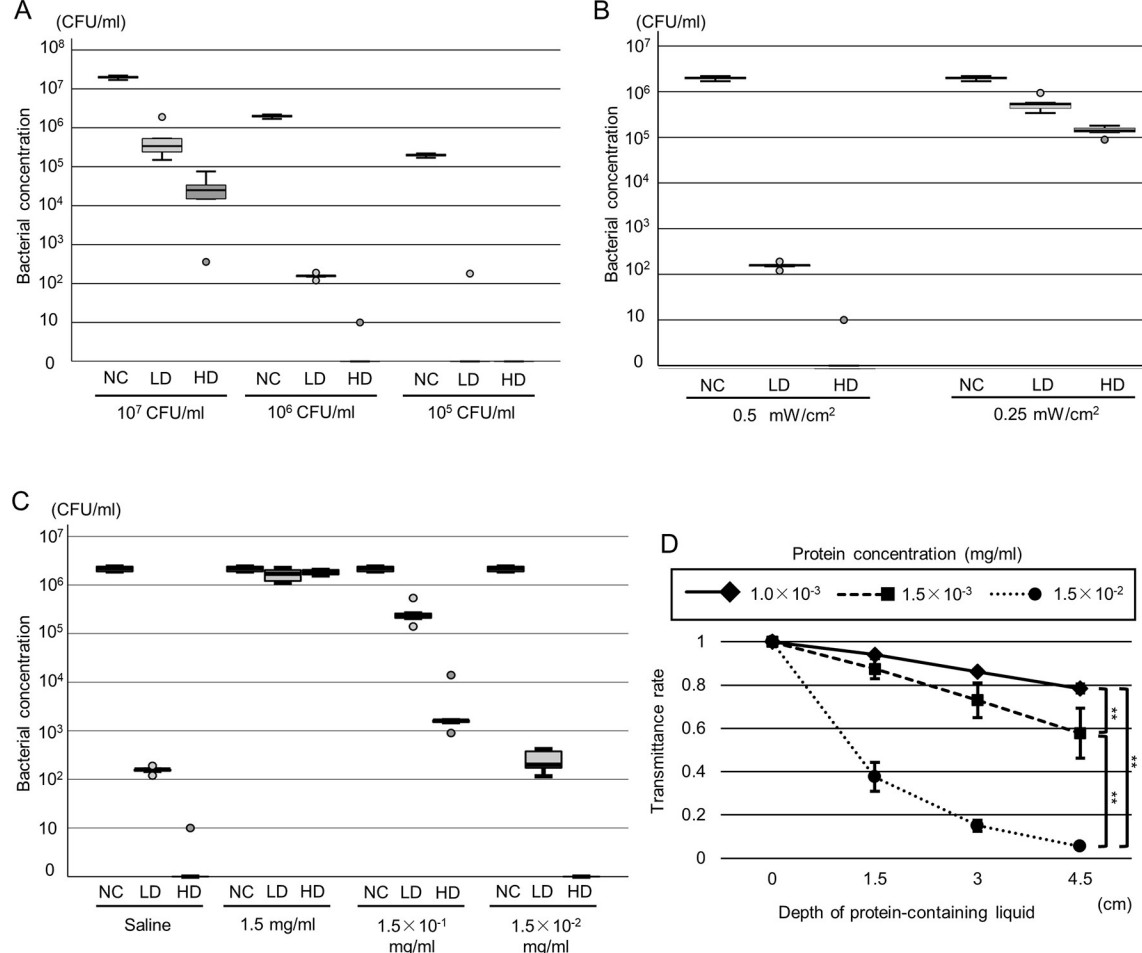

**Fig 2. Bactericidal effect of far UV-C irradiation at 222 nm *In Vitro*.** (A-C) Comparison of the bacterial concentration among the normal control (NC), low-dose (LD), and high-dose (HD) groups in different conditions. (A) Assessment of different bacterial concentrations: $1.0 \times 10^7$, $1.0 \times 10^6$, and $1.0 \times 10^5$ CFU/mL (n = 5 per group). (B) Assessment of different irradiance conditions: 0.5 and 0.25 mW/cm$^2$ (n = 5 per group). (C) Assessment of different protein concentrations. *E. coli* was diluted with saline or soy broth solution in concentrations of 1.5, $1.5 \times 10^{-1}$, and $1.5 \times 10^{-2}$ mg/mL (n = 5 per group). (D) The transmittance rate of far UV-C irradiation at 222 nm in the protein-containing solution at different concentrations: $1.0 \times 10^{-3}$, $1.5 \times 10^{-3}$, and $1.5 \times 10^{-2}$ mg/ml (n = 5 per group). **$p < 0.01$ (post hoc analysis with Tukey's procedure).

CFU/mL [$1.8 \times 10^5$–$4.0 \times 10^5$], HD: $1.6 \times 10^3$ CFU/mL [$1.2 \times 10^3$–$7.8 \times 10^3$]. At a concentration of $1.5 \times 10^{-2}$ mg/mL, the results were: NC: $2.0 \times 10^6$ CFU/mL [$1.9 \times 10^6$–$2.4 \times 10^6$], LD: $2.0 \times 10^2$ CFU/mL [$1.4 \times 10^2$–$4.0 \times 10^2$], HD: 0 CFU/mL [0–0], with a significance of $p < 0.01$. At a soy broth concentration of $1.5 \times 10^{-1}$ mg/mL, the bacterial concentrations showed a reduction of < 2 log CFU/mL between the NC and LD groups and between the LD and HD groups, while at concentrations of 0 (saline) and $1.5 \times 10^{-2}$ mg/mL, reductions of > 2 log CFU/mL were observed between these groups.

We evaluated the effect of protein concentration and solution depth on the transmittance rate of far UV-C irradiance at 222 nm (Fig 2D). The concentrations of the soy broth solution were adjusted to $1.0 \times 10^{-3}$, $1.5 \times 10^{-3}$, and $1.5 \times 10^{-2}$ mg/mL. The irradiance of far UV-C at 222 nm was set at 0.5 mW/cm$^2$. The transmittance rate was significantly different among the different soy broth concentrations (transmittance rate of $1.5 \times 10^{-2}$ mg/ml, 1.5 cm: $0.343 \pm 0.068$, 3.0 cm: $0.140 \pm 0.025$, 4.5 cm: $0.0522 \pm 0.0067$; transmittance rate of $1.5 \times 10^{-3}$

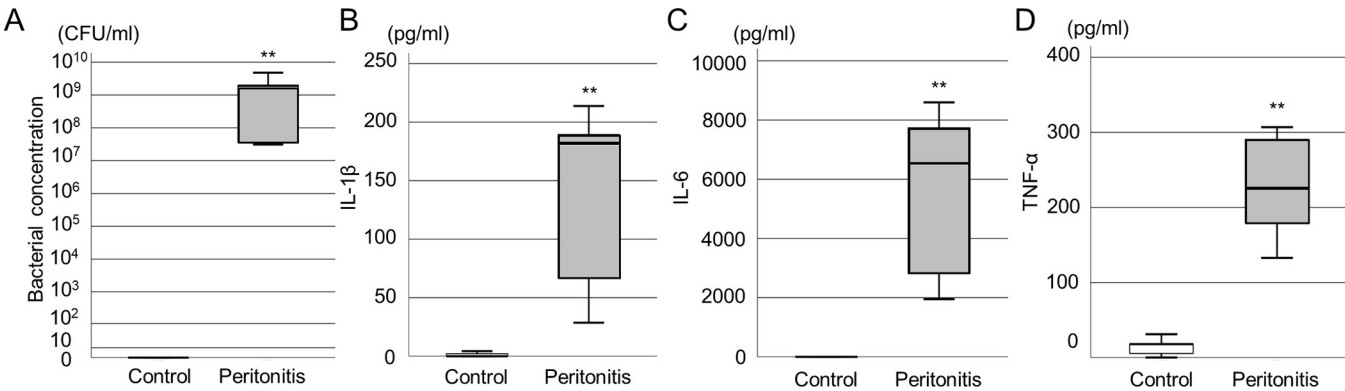

**Fig 3. Evaluation of ascites and serum cytokine levels in the peritonitis rat model.** (A) Bacterial concentration in ascites of the control and peritonitis groups (n = 5 per group). The bacterial concentration significantly increased in the peritonitis group. (B-D) Serum cytokine levels of the control and peritonitis groups (n = 5 per group). (B) interleukin (IL)-1β, (C) IL-6, and (D) tumor necrosis factor (TNF-α). The serum cytokine levels significantly increased in the peritonitis group. **$p < 0.01$ (Student's $t$-test).

mg/ml, 1.5 cm: 0.892 ± 0.046, 3.0 cm: 0.768 ± 0.080, 4.5 cm: 0.662 ± 0.116; transmittance rate of $1.0 \times 10^{-3}$ mg/ml, 1.5 cm: 0.944 ± 0.008, 3.0 cm: 0.866 ± 0.010, 4.5 cm: 0.793 ± 0.022; $p <$ 0.01). Significant differences in the transmittance rate were observed between concentrations of $1.0 \times 10^{-3}$ and $1.5 \times 10^{-2}$ mg/mL ($p < 0.01$) and those of $1.5 \times 10^{-3}$ and $1.5 \times 10^{-2}$ mg/mL ($p < 0.01$). At a depth of 4.5 cm, the transmittance rate was approximately 10% at a soy broth concentration of $1.5 \times 10^{-2}$ mg/mL. The transmittance rate was > 50% and significantly increased at soy broth concentrations of $1.0 \times 10^{-3}$ and $1.5 \times 10^{-3}$ mg/mL compared with that at $1.5 \times 10^{-2}$ mg/mL ($p < 0.01$). Based on these results, a bacterial concentration of $1.0 \times 10^{6}$ CFU/mL, an irradiance of 0.5 mW/cm$^2$ of far UV-C at 222 nm, and a protein concentration of $< 1.5 \times 10^{-3}$ mg/mL were desirable for the *in vivo* experiments.

## Evaluation of the rat peritonitis model

To confirm the induction of peritonitis in rats, bacteria in ascites and blood cytokines were evaluated 3 hours after *E. coli* administration. All rats in the control and peritonitis groups survived 3 hours after injection. Significantly higher concentrations of bacteria and serum cytokines were observed in the peritonitis group compared to the control group (Fig 3A–3D) ([bacterial concentration, control: 0 CFU/mL [0–0], peritonitis group: $1.9 \times 10^{9}$ CFU/mL [$3.4 \times 10^{7}$–$4.8 \times 10^{9}$], p < 0.01], [IL-1β, control: 0 pg/mL [0–3.26], peritonitis group: 183.0 pg/mL [47.6–201.5], p < 0.01], [IL-6, control: 0 pg/mL [0–0], peritonitis group: 6649.3 pg/mL [2381.5–8168.1], p < 0.01],] TNF-α, control: 17.9 pg/mL [2.8–24.5], peritonitis group: 226.3 pg/mL [155.9–298.8], p < 0.01]).

## Bactericidal effect of far UV-C irradiation on peritonitis in the acute phase

In a rat peritonitis model, we evaluated survival rate and DNA damage for the organ in the irradiation area as the effects of far UV-C irradiation at 222nm UV-C during lavage. The 1-week survival rate was significantly higher in the irradiated groups than in the non-irradiated group ($p = 0.038$, 60% vs. 20%) (Fig 4A). The bacterial concentration in the lavage solution was significantly decreased in the irradiated group than in the non-irradiated group (irradiated: $1.5 \times 10^{3}$ CFU/mL [$2.0 \times 10^{2}$–$7.9 \times 10^{4}$], non-irradiated: $2.5 \times 10^{5}$ CFU/mL [$1.7 \times 10^{5}$–$2.3 \times 10^{6}$], $p < 0.01$) (Fig 4B). The protein concentrations in the lavage solution were $< 2.0 \times 10^{-3}$ mg/mL in both groups, with no significant difference between them.

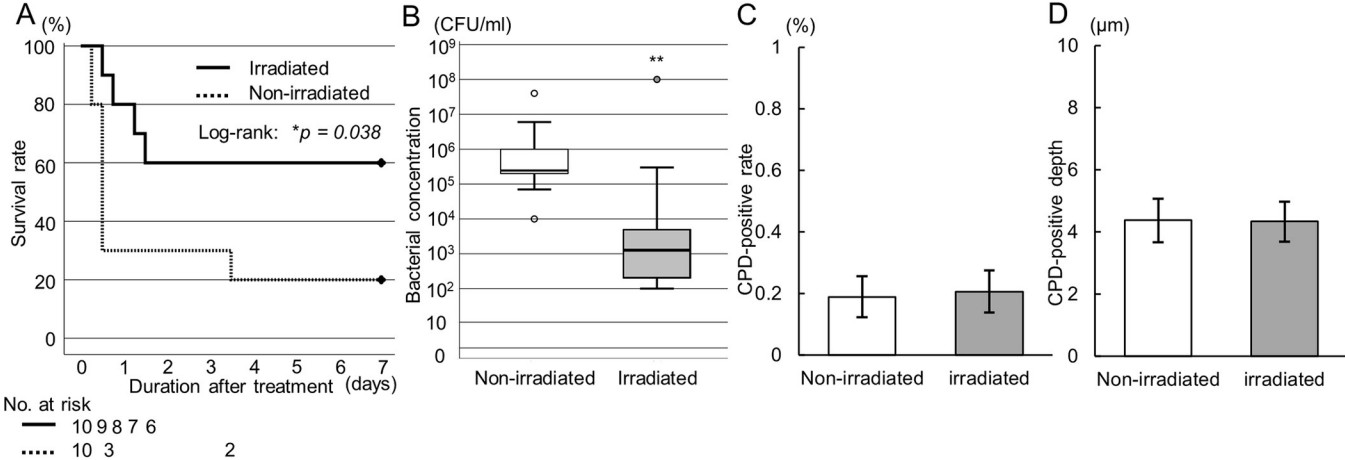

**Fig 4. Evaluation of the bactericidal effect of far UV-C irradiation on the rat peritonitis model in the acute phase.** (A) Survival rate during the 7 days after each procedure in the irradiated and non-irradiated groups (n = 10 per group, p = 0.038, log-rank test). (B) Bacterial concentration in ascites of the non-irradiated and irradiated groups (n = 10 per group). **$p < 0.01$ (Mann–Whitney $U$ test). (C) CPD-positive ratio and (D) depth in the cecum obtained on day 7 (non-irradiated, n = 2; irradiated group: n = 6).

Bacterial concentrations in peritoneal ascites were less than 10% dilution per lavage. Protein concentrations averaged less than 0.02 mg/dl in one lavage and 0.002 mg/dl in the second, below the minimum detection limit. Both the bacterial and protein concentrations met the conditions for the *in vitro* experiments.

The cecum was obtained from the surviving rats on day 7; of the surviving rats, two were in the non-irradiated group, and six were in the irradiated group. The CPD-positive rate was 0.21% ± 0.068% in the irradiated group and 0.19% ± 0.067% in the non-irradiated group ($p = 0.636$) (Fig 4C). The CPD-positive depth was 4.34 ± 0.64 μm in the irradiated group and 4.37 ± 0.70 μm in the non-irradiated group ($p = 0.920$) (Fig 4D). There were no significant differences in the CPD-positive rate or depth.

## Bactericidal effect of far UV-C irradiation on peritonitis in the hyper-acute phase

An in vivo study at the hyper-acute phase was conducted to evaluate the influence of far UV-C irradiation on blood cytokine level and bacterial concentration in ascites. In the peritonitis group, the bacterial concentrations significantly decreased in the irradiated group compared with the non-irradiated group (irradiated: $4.1 \times 10^2$ CFU/mL [$1.3 \times 10^2$–$1.1 \times 10^3$], non-irradiated: $1.3 \times 10^5$ CFU/mL [$1.2 \times 10^4$–$5.8 \times 10^5$], $p < 0.01$) (Fig 5A).

The serum IL-1β and IL-6 levels significantly decreased in the irradiated peritonitis group compared with the non-irradiated group (IL-1β; irradiated: 68.7 pg/mL [40.4–87.6], non-irradiated: 109.5 pg/mL [92.4–149.2], $p < 0.01$; IL-6; irradiated: 1,367.6 pg/mL [751.0–1,731.0], non-irradiated: 3458.2 pg/mL [2,374.3–4,973.8], $p = 0.01$) (Fig 5B, 5C). In the sham group, no significant differences in the bacterial concentration and serum IL-1β and IL-6 levels between the irradiated and non-irradiated groups were observed (bacterial concentration; irradiated: 0 CFU/mL [0–0], non-irradiated: 0 CFU/mL [0–0], $p = 1.000$) (IL-1β; irradiated: 0 pg/mL [0–4.4], non-irradiated: 0 pg/mL [0–11.1], $p = 0.841$) (IL-6; irradiated: 0 pg/mL [0–0], non-irradiated: 0 pg/mL [0–55.6], $p = 0.690$) (Fig 5A–5C). Further, both in the sham and peritonitis groups, there were no significant differences in the serum TNF-α levels between the irradiated and non-irradiated groups (peritonitis group; irradiated: 73.3 pg/mL [58.9–146.3], non-

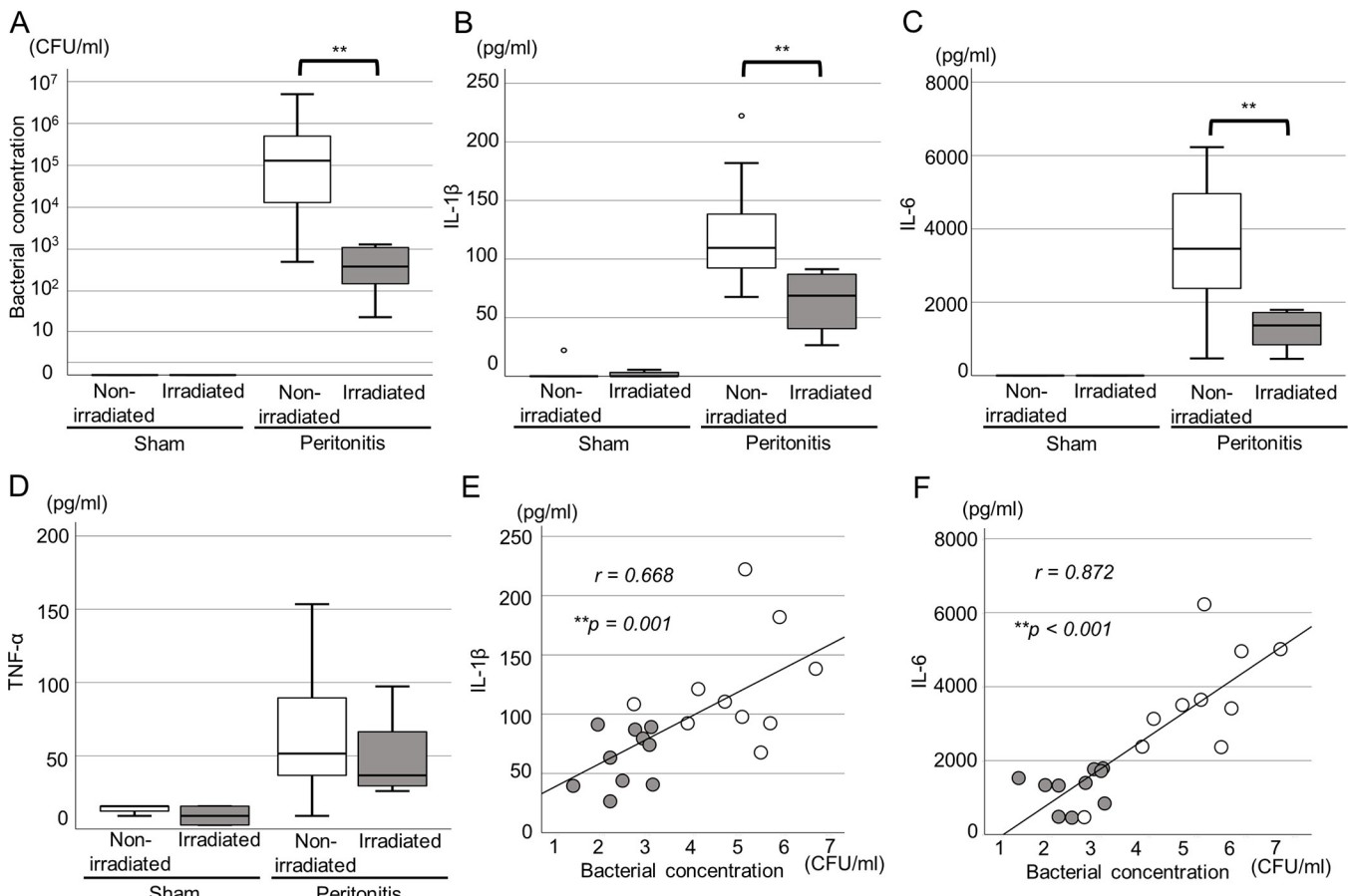

**Fig 5. Evaluation of the bactericidal effect of far UV-C irradiation for the peritonitis rat model in the hyper-acute phase.** Bacterial concentration in ascites (A) and serum interleukin (IL)-1β (B), IL-6 (C), and tumor necrosis factor (TNF-α) (D) levels in the sham (non-irradiated, n = 5, irradiated, n = 5) and peritonitis groups (non-irradiated, n = 10, irradiated, n = 10). The bacterial concentration and serum IL-1β and IL-6 levels significantly decreased in the irradiated peritonitis group (**$p < 0.01$, Student's t-test). Correlations between the bacterial concentration in ascites and serum IL-1β level (E) ($r = 0.644$, $p < 0.01$) and that between the bacterial concentration in ascites and serum IL-6 level (F) ($r = 0.838$, $p < 0.01$) in the peritonitis group. Gray circles indicate the irradiated group and white circles indicate the non-irradiated group.

irradiated: 103.0 pg/mL [69.7–194.8], $p = 0.436$; sham group; irradiated: 17.9 pg/mL [5.5–31.1], non-irradiated: 31.1 pg/mL [21.2–31.1], $p = 0.310$) (Fig 5D).

The relationship between the bacterial concentration and serum IL-1β and IL-6 levels is shown in Fig 5E and 5F; a positive correlation was noted (bacterial concentration and serum IL-1β level; $r = 0.644$, 95% confidence interval [CI], 0.266–0.841, $p < 0.01$; bacterial concentration and serum IL-6 level; $r = 0.838$, 95% CI, 0.615–0.931, $p < 0.01$).

## DNA damage in the abdominal tissues by far UV-C irradiation

We evaluated DNA damage to intraperitoneal organs by far UV-C irradiation to confirm the safety of this procedure. IHC for CPDs was performed to evaluate the DNA damage induced by UV-C irradiation (Fig 6A). CPD-positive cells were not observed in any tissue in the NC group, while only a few CPD-positive cells were observed in the serosa of abdominal tissues in the LD and HD groups. Conversely, there were no CPD-positive cells in the internal tissues. In the PC group, many CPD-positive cells were observed in both the serosa and internal tissues. The CPD-positive rate was significantly lower for all tissues in the LD and HD groups than for

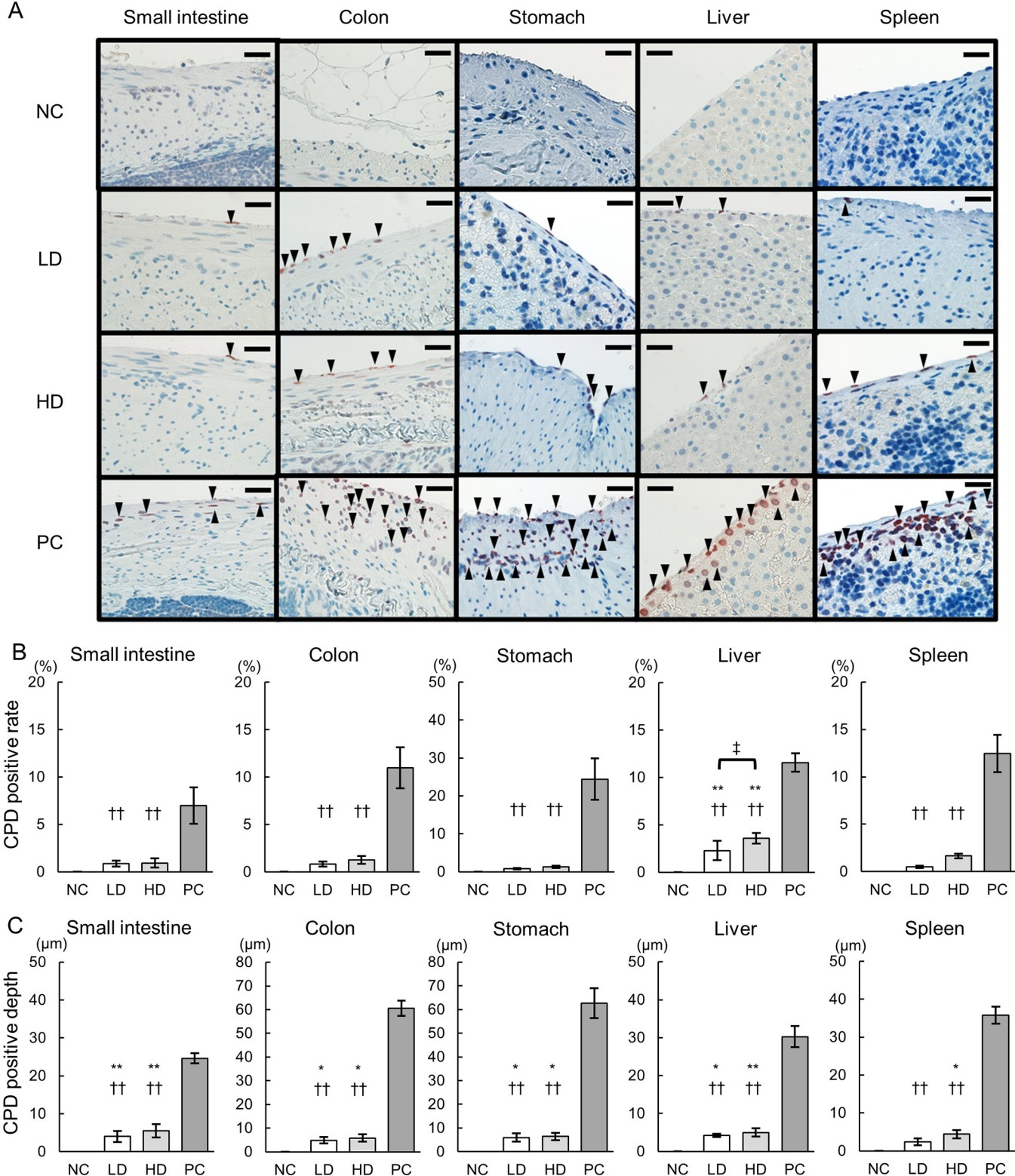

**Fig 6. Cytotoxicity in the abdominal tissues by UV-C irradiation.** (A) Immunohistochemical (IHC) staining for CPDs in the irradiated abdominal tissues (NC, negative control; LD, low-dose; HD, high-dose; PC, positive control). Arrowheads indicate CPD-positive cells. Scale bars = 20 μm. The CPD-positive rate (B) and CPD-positive depth (C) in the irradiated abdominal tissues are shown (n = 5 per group). **p < 0.01 and *p < 0.05 indicate statistical significance compared with the NC group, while ††p < 0.01 denotes significance compared with the PC group, ‡p < 0.05. Statistical analysis was performed using ANOVA followed by Tukey's post hoc procedure.

those in the PC groups (CPD-positive rates of the small intestine, NC: 0% ± 0, LD: 0.92% ± 0.31, HD: 1.10% ± 0.48, PC: 6.97% ± 1.90, CPD-positive rates of the colon, NC: 0% ± 0, LD: 0.83% ± 0.28, HD: 1.27% ± 0.39, PC: 10.99% ± 2.17, CPD-positive rates of the stomach, NC: 0% ± 0, LD: 0.75% ± 0.16, HD: 1.24% ± 0.28, PC: 22.27% ± 5.45, CPD-positive rates of the liver, NC: 0% ± 0, LD: 2.44% ± 1.00, HD: 3.84% ± 0.54, PC: 11.55% ± 2.83, CPD-positive rates of the spleen, NC: 0% ± 0, LD: 0.45% ± 0.10, HD: 1.74% ± 0.22, PC: 12.00% ± 1.97) (Fig 6B). There was no significant difference in the CPD-positive rate in the small intestine, ascending colon, stomach, or spleen between the LD and NC groups or between the HD and NC groups. However, there was a significant difference in the CPD-positive rate in the liver alone between the LD and NC groups and between the HD and NC groups.

The CPD-positive depth was significantly lower in all tissues in the LD and HD groups than in those in the PC group. Furthermore, it was significantly higher in all tissues, except for the spleen, in the LD group compared to the NC group. Specifically, the CPD-positive depths in various tissues were as follows: small intestine (NC: 0 µm ± 0, LD: 4.31 µm ± 1.47, HD: 5.44 µm ± 1.78, PC: 24.92 µm ± 1.36), colon (NC: 0 µm ± 0, LD: 5.05 µm ± 1.36, HD: 5.67 µm ± 1.59, PC: 60.28 µm ± 3.25), stomach (NC: 0 µm ± 0, LD: 5.69 µm ± 1.76, HD: 6.06 µm ± 1.54, PC: 61.03 µm ± 6.38), liver (NC: 0 µm ± 0, LD: 4.34 µm ± 0.42, HD: 5.15 µm ± 1.05, PC: 31.45 µm ± 1.05), and spleen (NC: 0 µm ± 0, LD: 2.80 µm ± 0.94, HD: 4.13 µm ± 1.07, PC: 36.21 µm ± 2.28) (Fig 6C).

## Discussion

This study optimized far-UVC irradiation at 222 nm for its bactericidal effects and cytotoxicity in acute peritonitis. *In vitro* experiments showed significant reductions in bacterial concentrations by far-UVC irradiation at 222 nm. The bactericidal effects varied depending on bacterial concentration, protein concentration, and the depth of the liquid. These results suggested that it is necessary to irradiate under appropriate conditions to achieve the bactericidal effects of far-UVC irradiation at 222 nm. In vivo, far-UVC irradiation at 222 nm under appropriate conditions significantly improved survival rates, reduced bacterial concentrations in lavage solutions, and improved serum inflammatory cytokine levels. Minimal DNA damage was observed in abdominal tissues, confirming safety.

Ascites caused by peritonitis contains a high concentration of proteins and biochemical substances. Far UV-C irradiance at 222 nm is strongly absorbed by proteins, which limits its penetration ability in ascites with high protein concentrations [35, 36]. In our study, the bactericidal effects of far-UVC irradiation at 222 nm were insufficient in solutions with high protein concentrations. In contrast, at a protein concentration of $1.5 \times 10^{-2}$ mg/mL, the bactericidal effects were similar to those of saline. Based on these results, it is considered necessary to reduce the protein concentration through intraperitoneal lavage to achieve sufficient bactericidal effects with far-UVC irradiation at 222 nm. Furthermore, the depth of the solution affects the transmittance rate of UVC and influences the bactericidal effect. In our *in vitro* experiments, higher protein concentrations resulted in greater attenuation of 222 nm UVC.

Furthermore, ascites resulting from bacterial peritonitis typically exhibits a high concentration of bacteria. This bacterial concentration also influences the bactericidal effects of far-UVC irradiation [40]. Notably, *E. coli* densities exceeding $10^7$ CFU/mL lead to bacterial aggregation. Because far-UVC irradiation cannot penetrate these aggregates, the bactericidal effect is compromised [40]. In our *in vitro* experiments, the bactericidal effect was insufficient at $10^7$ CFU/mL, yet it proved adequate at $10^5$ CFU/mL. Consequently, it is deemed essential to lower the bacterial concentration and avert bacterial aggregation via intraperitoneal agitated lavage to ensure effective bactericidal outcomes with far-UVC irradiation at 222 nm.

From these *in vitro* experiments, it was determined that achieving adequate bactericidal effects with UVC at 222 nm necessitates adjusting both protein and bacterial concentrations in the ascites to suitable levels. In our peritonitis model, the bacterial concentration in the ascites and serum cytokine levels escalated, simulating the condition seen in human peritonitis. Moreover, lavage with saline decreased both bacterial and protein concentrations in the lavage solutions, thereby enhancing the bactericidal efficacy of far-UVC irradiation at 222 nm. The bactericidal impact of far-UVC irradiation at 222 nm was assessed based on bacterial reduction in ascites, systemic inflammation as indicated by cytokines, and survival rates. In our in vivo experiments, far-UVC irradiation at 222 nm further diminished the bacterial concentration in the ascites and improved survival rates compared to lavage alone. In acute peritonitis, the association between inflammatory cytokines and mortality rates has been reported. Mainly, high IL-1β and IL-6 serum levels are associated with a high mortality risk in acute peritonitis [41–43]. In our in vivo study, far-UVC irradiation at 222 nm reduced the bacterial concentration in the lavage solution and serum levels of IL-1β and IL-6. Additionally, there was a correlation between the bacterial concentration and serum levels of IL-1β and IL-6. These findings suggested that the bactericidal effect of far-UVC irradiation at 222 nm might also improve serum inflammatory cytokine concentrations, indicating its potential to treat peritonitis by reducing the release of inflammatory cytokines.

Confirming the safety of far UV-C irradiation at 222 nm on intra-abdominal organs was another challenge in this study. UV-C irradiation at both 222 nm and 254 nm exerts strong bactericidal effects through DNA damage, which represses transcription and replication, leading to cell death [32]. This DNA damage occurs not only in bacteria but also in mammalian cells through similar mechanisms. In our in vivo experiments, DNA damage due to UV-C irradiation was assessed by evaluating both the CPD-positive ratio and depth. The CPD-positive rate and depth were significantly lower at 222 nm compared to 254 nm across all organs. UV-C irradiation at 222 nm is more readily absorbed by cytoplasmic proteins than at 254 nm [32]. Owing to these absorption properties, only about 50% of 222 nm irradiation penetrates less than 1 μm into cells, versus 3 μm for 254 nm irradiation [33, 34]. Mammalian cell diameters range from 10–25 μm, whereas those of bacteria are less than 1 μm [44, 45]. Therefore, far UV-C irradiation at 222 nm could effectively kill bacteria, and due to its limited penetration into mammalian cells, provide safety advantages over 254 nm irradiation.

In the subsequent phase, we explored the safety of 222 nm UV-C irradiation in vivo. No difference in CPD-positive ratio or CPD-positive depth was observed between the group exposed to a low dose of 222 nm UV-C and the non-irradiated group. Consequently, exposure to 222 nm UV-C at a low dose appears to be safe in the short term. When comparing low and high doses of 222 nm UV-C exposure, the CPD-positive ratio increased significantly only at high doses in liver tissues, demonstrating a dose-dependent increase in cytotoxicity. However, the CPD-positive depth remained below 10 μm in both groups, with no significant difference noted. No CPD-positive cells were detected in the internal tissues of any organs in either group. These findings indicate that in the high-dose group, which was exposed for up to 1000 s, the damage is confined to the organ serosa, suggesting that the internal tissues remained undamaged. However, damage confined to the serosa does not guarantee long-term safety. Thus, prolonged follow-up studies are essential to assess the long-term safety of 222 nm UV-C irradiation.

The bactericidal effect of 222 nm UV-C radiation on *E. coli* is attributed to DNA damage [46]. However, other photoinactivation mechanisms by 222 nm UV-C radiation against bacteria remain unclear. In future studies, we will focus on its therapeutic effects against lipopolysaccharide (LPS), a component of the outer membrane of Gram-negative bacteria. LPS plays a crucial role in pathogenesis, triggers a robust immune response, and induces the release of

inflammatory cytokines [41–43]. In this study, far-UVC irradiation at 222 nm improved serum inflammatory cytokine levels, prompting speculation about a potential relationship between LPS and 222 nm UV-C radiation [47–51]. Therefore, upcoming research aims to investigate the therapeutic effects of 222 nm UV-C against the LPS-induced inflammatory pathway.

Far UV-C irradiation at 222 nm has demonstrated bactericidal effects against various bacteria, viruses, and fungi [26, 27, 46, 52]. Future research aims to investigate the bactericidal effects of 222 nm UV-C irradiation on pathogens other than *E. coli*. Recently, infections caused by antibiotic-resistant bacteria have become a significant concern [53]. The bactericidal effects of UV-C against antibiotic-resistant bacteria may offer therapeutic benefits throughout numerous infections.

## Conclusion

This study demonstrated that far UV-C irradiation at 222 nm, applied at appropriate bacterial and protein concentrations and irradiance levels, exerts significant bactericidal effects both in vivo and *in vitro*. DNA damage to abdominal organs was minimal with low-dose 222 nm UV-C irradiation, suggesting that this treatment has a favorable safety profile. Therefore, far UV-C irradiation at 222 nm shows potential as an effective treatment for peritonitis and warrants further research for clinical application.

## Acknowledgments

The authors would like to thank Editage (www.editage.jp) for English language editing.

## Author Contributions

**Conceptualization:** Kosuke Sugiyama, Kiyotaka Kurachi, Hiroya Takeuchi.

**Data curation:** Kosuke Sugiyama, Kiyotaka Kurachi, Kyota Tatsuta, Tomoaki Fukui.

**Formal analysis:** Kosuke Sugiyama, Kiyotaka Kurachi, Masaki Sano, Tadahiro Kojima, Toshiya Akai, Katsunori Suzuki, Kakeru Torii, Mayu Sakata, Yoshifumi Morita, Hirotoshi Kikuchi, Yoshihiro Hiramatsu, Hiroya Takeuchi.

**Funding acquisition:** Kiyotaka Kurachi, Mayu Sakata, Hiroya Takeuchi.

**Investigation:** Kosuke Sugiyama.

**Methodology:** Kosuke Sugiyama, Kiyotaka Kurachi, Masaki Sano, Kyota Tatsuta, Tadahiro Kojima, Toshiya Akai, Katsunori Suzuki, Kakeru Torii, Mayu Sakata, Yoshifumi Morita, Hirotoshi Kikuchi, Yoshihiro Hiramatsu, Tomoaki Fukui, Rena Kaigome, Masahiro Sasaki, Toru Koi, Tetsuro Suzuki, Ryosuke Kuroda, Hiroya Takeuchi.

**Project administration:** Kosuke Sugiyama, Kiyotaka Kurachi, Tomoaki Fukui, Toru Koi, Ryosuke Kuroda, Hiroya Takeuchi.

**Resources:** Rena Kaigome, Masahiro Sasaki, Toru Koi, Hiroyuki Ohashi, Tetsuro Suzuki.

**Supervision:** Toru Koi, Ryosuke Kuroda, Hiroya Takeuchi.

**Validation:** Rena Kaigome, Masahiro Sasaki, Toru Koi, Hiroyuki Ohashi.

**Visualization:** Kosuke Sugiyama.

**Writing – original draft:** Kosuke Sugiyama, Masaki Sano.

**Writing – review & editing:** Kosuke Sugiyama, Kiyotaka Kurachi, Masaki Sano, Kyota Tat-suta, Tadahiro Kojima, Toshiya Akai, Katsunori Suzuki, Kakeru Torii, Mayu Sakata, Yoshi-fumi Morita, Hirotoshi Kikuchi, Yoshihiro Hiramatsu, Yohei Kumabe, Keisuke Oe, Tomoaki Fukui, Rena Kaigome, Masahiro Sasaki, Toru Koi, Hiroyuki Ohashi, Tetsuro Suzuki, Ryosuke Kuroda, Hiroya Takeuchi.

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
