## [Decision Letter · Decision Letter 0]

13 May 2024

PONE-D-24-00293

Bactericidal effect of far ultraviolet-C irradiation at 222 nm against bacterial peritonitisPLOS ONE

Dear Dr. Kurachi,

Thank you for submitting your manuscript to PLOS ONE. After careful consideration, we feel that it has merit but does not fully meet PLOS ONE’s publication criteria as it currently stands. Therefore, we invite you to submit a revised version of the manuscript that addresses the points raised during the review process.

We look forward to receiving your revised manuscript.

Kind regards,

Seyed Mostafa Hosseini

Academic Editor

PLOS ONE

Journal Requirements:

[Because this research required specialized experimental equipment, it was started as a joint research project between Kobe University, Ushio Inc., and Hamamatsu University of Medicine. This experiment was performed at Hamamatsu University School of Medicine. The staff of Ushio Inc. and Kobe University lent specialized equipment used in another study to the Hamamatsu University School of Medicine, evaluated its appropriate use, assessed the validity of the methods and results, and confirmed the reproducibility of some data. Thus, the funders had no role in data collection and analysis. Based on the result of this research, Hamamatsu University School of Medicine have received funds from Ushio Inc. associated with this collaborative project. Kobe University School of Medicine and Hamamatsu University School of Medicine are funded by Ushio Inc. for collaborating research. RK, MS, TK, and HO have received support in the form of salaries from Ushio Inc.]. 

Reviewers' comments:

Reviewer's Responses to Questions

**Comments to the Author**

1. Is the manuscript technically sound, and do the data support the conclusions?

Reviewer #1: Partly

Reviewer #2: Yes

2. Has the statistical analysis been performed appropriately and rigorously? 

Reviewer #1: Yes

Reviewer #2: Yes

3. Have the authors made all data underlying the findings in their manuscript fully available?

Reviewer #1: Yes

Reviewer #2: Yes

4. Is the manuscript presented in an intelligible fashion and written in standard English?

Reviewer #1: Yes

Reviewer #2: No

5. Review Comments to the Author

Reviewer #1: Dear Authors

Thanks for your manuscript with title: Bactericidal effect of far ultraviolet-C irradiation at 222 nm against bacterial peritonitis. The review of the aforementioned manuscript has been finished despite its interesting, there are some issues about it which you could find at the attached file.

Best Regards

Reviewer #2: 1- The text of the article has many problems in terms of writing and grammar.Edite the article in terms of grammar and writing.

2- What type of laser was used?

3- Explain the reason for using the wavelength of 222 and the time of irradiation?

4- There is no explanation about the distance from the laser to the peritonitis?

5- Why are variable wavelengths not used?

6- Why was antibiotics not used as a control?

6. PLOS authors have the option to publish the peer review history of their article (what does this mean?). If published, this will include your full peer review and any attached files.

Reviewer #1: No

Reviewer #2: No

---

## [Author Response · Author response to Decision Letter 0]

11 Sep 2024

Thank you for checking our paper and for your numerous comments.

We have carefully considered each comment and responded to them.

We look forward to hearing from you in good faith.

---

## [Decision Letter · Decision Letter 1]

23 Sep 2024

Bactericidal effect of far ultraviolet-C irradiation at 222 nm against bacterial peritonitis

PONE-D-24-00293R1

Dear Dr. Kurachi,

We’re pleased to inform you that your manuscript has been judged scientifically suitable for publication and will be formally accepted for publication once it meets all outstanding technical requirements.

Kind regards,

Geelsu Hwang, Ph.D.

Academic Editor

PLOS ONE

Reviewers' comments:

Reviewer's Responses to Questions

**Comments to the Author**

1. If the authors have adequately addressed your comments raised in a previous round of review and you feel that this manuscript is now acceptable for publication, you may indicate that here to bypass the “Comments to the Author” section, enter your conflict of interest statement in the “Confidential to Editor” section, and submit your "Accept" recommendation.

Reviewer #1: All comments have been addressed

Reviewer #2: All comments have been addressed

2. Is the manuscript technically sound, and do the data support the conclusions?

Reviewer #1: Yes

Reviewer #2: Yes

3. Has the statistical analysis been performed appropriately and rigorously? 

Reviewer #1: Yes

Reviewer #2: Yes

4. Have the authors made all data underlying the findings in their manuscript fully available?

Reviewer #1: Yes

Reviewer #2: Yes

5. Is the manuscript presented in an intelligible fashion and written in standard English?

Reviewer #1: Yes

Reviewer #2: Yes

6. Review Comments to the Author

Reviewer #1: Dear Authors

Thanks for your revised manuscript with title: Bactericidal effect of far ultraviolet-C irradiation at 222 nm against bacterial peritonitis. The review of the revised manuscript has been finished and the authors addressed almost all the issues.

Best Regards

Reviewer #2: (No Response)

7. PLOS authors have the option to publish the peer review history of their article (what does this mean?). If published, this will include your full peer review and any attached files.

Reviewer #1: No

Reviewer #2: No

---

## [Editor Report · Acceptance letter]

4 Oct 2024

PONE-D-24-00293R1 

PLOS ONE

Dear Dr. Kurachi, 

I'm pleased to inform you that your manuscript has been deemed suitable for publication in PLOS ONE. Congratulations! Your manuscript is now being handed over to our production team.

Kind regards, 

on behalf of

Dr. Geelsu Hwang 

Academic Editor

PLOS ONE